# Spices and Atherosclerosis

**DOI:** 10.3390/nu10111724

**Published:** 2018-11-10

**Authors:** Pi-Fen Tsui, Chin-Sheng Lin, Ling-Jun Ho, Jenn-Haung Lai

**Affiliations:** 1Graduate Institute of Life Sciences, National Defense Medical Center, Taipei 11490, Taiwan; befun1214@gmail.com; 2Division of Cardiology, Department of Medicine, Tri-Service General Hospital, National Defense Medical Center, Taipei 11490, Taiwan; littlelincs@gmail.com; 3Institute of Cellular and System Medicine, National Health Research Institute, Zhunan 35053, Taiwan; lingjunho@nhri.org.tw; 4Division of Allergy, Immunology and Rheumatology, Department of Internal Medicine, Chang Gung Memorial Hospital, Chang Gung University, Tao-Yuan 33305, Taiwan

**Keywords:** atherosclerosis, atherosclerotic cardiovascular disease, spices, nutrients

## Abstract

Cardiovascular disease is one of the leading causes of death and disability in the world. Atherosclerosis, characterized by lipid accumulation and chronic inflammation in the vessel wall, is the main feature of cardiovascular disease. Although the amounts of fruits and vegetables present in the diets vary by country, diets, worldwide, contain large amounts of spices; this may have positive or negative effects on the initiation and development of atherosclerosis. In this review, we focused on the potential protective effects of specific nutrients from spices, such as pepper, ginger, garlic, onion, cinnamon and chili, in atherosclerosis and atherosclerotic cardiovascular disease. The mechanisms, epidemiological analysis, and clinical studies focusing on a variety of spices are covered in this review. Based on the integrated information, we aimed to raise specific recommendations for people with different dietary styles for the prevention of atherosclerotic cardiovascular disease through dietary habit adjustments.

## 1. Introduction

### 1.1. Diet and Cardiovascular Disease

Cardiovascular disease (CVD) is one of the leading causes of death worldwide, which may include coronary artery disease, acute myocardial infarction, peripheral arterial disease and stroke. The prevalence of CVD is estimated to increase from 36.9% to 40.5% from 2010 to 2030 in the United States, and the associated medical cost will increase by 200% [1]. Therefore, it is vital to conduct biomedical research studies focusing on effective prevention and treatment strategies for CVD. Atherosclerosis is the main cause of CVDs. The risk factors of atherosclerosis and CVD include diabetes, smoking, hypertension, dyslipidemia, obesity and age [2]. Moreover, nutritionists have reported that the intake of a diet with lower proportions of fat, sugar and salt, but high amounts of various vegetables and fruits may effectively reduce the incidence of CVD and obesity. However, eating habits and food consumptions patterns vary across cultures and religions.

The typical western diet comprises high levels of red meat, animal fat and sugar, and low levels of fruits and vegetables. The Mediterranean diet appears to be relatively healthy, as it includes vegetables, fruits, fish, whole grains, beans and olive oil. People living in southern Europe, along the Mediterranean coast, in countries such as Greece, Spain, France and southern Italy typically eat a Mediterranean diet. Herbs and spices are commonly used as adjuvants in Mediterranean-style cooking to increase the flavor of food and reduce the need for salt and oil [3]. People taking a Mediterranean diet have a low incidence of CVD, diabetes, and hyperlipidaemia [4]. Indeed, a 10-year follow-up with a self-administered questionnaire showed that the intake of a Mediterranean diet reduces the incidence of CVD and prevents atherosclerosis [5]. A multi-centre clinical trial in Spain showed that the intake of a Mediterranean diet with extra-virgin olive oil or nuts lowered the risk of CVD events compared to a conventional low-fat diet in participants who had a high CVD risk [6].

In China, the diet is predominantly plant-based, which is beneficial to CVD prevention compared to omnivorous diets; however, the cooking style—characterized by the addition of high amounts of salt and sugar—may increase the CVD prevalence in that country [7,8]. Indians also consume a predominantly plant-based diet, which includes a variety of spices that have possible anti-hypercholesterolemic and anti-inflammatory effects [9]. The diet of people in Arab countries includes fruits, vegetables, meat, milk and many spices. During festivals such as Eidal-Adha, Arabians consume large amounts of meat and sugary sweets, which may contribute to the development of CVD. Although Islamic fasting has been proven to prevent CVD by improving the metabolic process of diabetes, CVD and metabolic syndrome are still highly prevalent in Muslim countries [10]. Such evidence highlights the important link between dietary style and CVD.

### 1.2. Atherosclerosis: A Critical Pathogenesis of CVD

Atherosclerosis is characterized by vascular inflammation and lipid accumulation within the vessel wall. It is initiated by endothelial cell (EC) damage and the attachment of oxidized low-density lipoprotein (oxLDL) on the inner wall of an artery. The inflammation of ECs leads to a reduction in the production of endothelial nitric oxide synthase (eNOS) and nitric oxide (NO) but induces the expression of adhesion molecules such as intercellular adhesion molecule 1 (ICAM-1) and vascular cell adhesion molecule 1 (VCAM-1); this further promotes monocyte adhesion and migration. Thereon, the uptake of oxLDL by macrophages occurs, followed by their differentiation into foam cells. The redundant cholesterol is then predominantly expelled by ATP-binding cassette transporter A1 (ABCA1) and ABCG1, delivered via apolipoprotein A-1 (ApoA-1) and high-density lipoprotein (HDL), respectively, and finally excreted through bile [11,12]. The expression of ABCA1 is regulated by peroxisome proliferator-activated receptors gamma (PPARγ)—a nuclear receptor—and liver X receptor alpha (LXRα) for the promotion of cholesterol efflux, which inversely correlates with CVD [13]. Meanwhile, the active LXRα represses nuclear factor-κB (NFκB) signaling, which provides protection against vascular inflammation [14]. The foam cells stimulate the ECs to produce cytokines and chemokines to attract further monocyte migration. This leads to the proliferation of vascular smooth muscle cells (VSMCs) and their migration to the intima. Other immune cells, including dendritic cells and T cells, could be activated by various stimuli, which promote plaque formation and the development of atherosclerosis [15]. During the process of atherosclerosis, reactive oxygen species (ROS) induce the oxidative injury of vessels, which then initiates atherogenesis. Additionally, ROS are known as second messengers in the initiation of inflammatory signaling, including mitogen-activated protein kinases (MAPKs) and NF-κB, which can be activated by several other inflammatory stimuli for the production of pro-inflammatory cytokines [16,17].

### 1.3. Spices and Atherosclerosis

Nutrition research suggests that the nutrients contained in vegetables and fruits offer a host of atheroprotective effects. For example, the intake of fibber—a plant-based nutrient—is associated with reduced CVD risk [18]. Flavonoids, which are abundantly present in plant-based foods, can induce cholesterol efflux and increase HDL cholesterol (HDL-C) concentrations [19]. However, the production of crops is influenced by local climate, temperature and environment. The types of spices consumed vary across cultures (Table 1). In the Mediterranean diet, herbs and spices are commonly used in cooking to increase the flavor of food and to reduce the need for salt and oil [3]. Five-spice powder is among the popular condiments in China, and includes peppercorns, fennel, cloves, star anise, and cinnamon, all of which have been proven to improve CVD symptoms. For example, Piper, the genus of pepper plants, had a preventive role in oxidative stress in the cardiac tissue of hamsters fed an atherogenic diet [20]. Furthermore, spices are cheaper and easier to transport than vegetables and fruits. In the following sections, we will discuss several spices that have atheroprotective effects, as well as the mechanisms involved in these effects.

## 2. Spices with Potential Atheroprotective Effects

### 2.1. Black Pepper (Piper nigrum)

Black pepper, native to south India, is widely cultivated in tropical regions such as Southeast Asia. It is commonly used with salt as seasoning, globally. Extracted pepper oil has medicinal value and can be used in beauty products. In traditional medicine, black pepper is used to improve digestion, potentiate antibacterial effects, and treat common cold.

Pepper extracts were found to have an antioxidant role in the cardiac, hepatic and renal tissues of hamsters fed a high-fat diet (HFD) [20]. Moreover, black pepper improves the lipid profile, including the levels of total cholesterol, LDL and triglycerides in rats that were fed cholesterol [25]. Although the beneficial role of black pepper on hyperlipidemia has been demonstrated in rats, the effects of black pepper on cardiometabolic disease as well as the amounts used in human need further studies to verify. The pepper extract, piperine, inhibits platelet-derived growth factor-BB-induced proliferation and migration in VSMCs via the downregulation of extracellular signal-regulated kinases (ERKs) and p38 activities [26] (Table 2, Figure 1A). Piperine suppresses platelet aggregation and the inflammatory reaction of macrophages by inhibiting the activities of cytosolic phospholipase A_2_ and COX-2, respectively [27]. In addition, piperine increases ABCA1 expression and promotes cholesterol efflux by inhibiting ABCA-1 degradation in the human monocytic line, THP-1 macrophages [28]. Through the regulation of lipid metabolism, inhibition of ROS production, and reduction in the rate of VSMC proliferation via the suppression of MAPK signaling, pepper offers therapeutic effects in atherosclerotic CVD.

### 2.2. Cinnamon (Cinnamomum)

Cinnamon is native to India and Sri Lanka. Currently, its cultivation is distributed across the low-latitude mountainous areas of China, Indonesia, Sri Lanka, India and Vietnam. The bark, branches, leaves, fruits and pedicels of cinnamon can be used in food or medicines. In Chinese medicine, cinnamon is used to promote gastrointestinal motility, improve digestion and inhibit bacterial growth. In patients with type 2 diabetes, short-term intake of 2 g of cinnamon has been proven to significantly reduce levels of hemoglobin A1C and blood pressure [29]. Cinnamon reversibly and competitively inhibits alpha-glucosidase enzyme and improves postprandial hyperglycemia in streptozotocin-induced diabetic rats [30]. The cinnamon extract—cinnamaldehyde—exerts anticoagulant effects in the prevention of unnecessary clumping of platelets [31]. It has excellent antioxidant capacity by using in vitro models, including the beta-carotene-linoleate and phosphomolybdenum complex methods [32].

Cinnamon was found to inhibit the atherosclerosis process by the prevention of apoA-1 glycation and inhibition of cholesteryl ester transfer protein (CETP) in hypercholesterolemic zebrafish [33]. Cinnamon extract treatment significantly reduced the rate of dyslipidaemia and aided in the maintenance of the atherogenic index (total cholesterol—HDL-C/HDL-C) compared to dexamethasone control in atherosclerotic rats [34]. Additionally, cinnamon aqueous extract inhibited the expressions of CD36 and scavenger receptor class A (SRA), as well as acetyl LDL uptake via the regulation of ERK1/2 activity in macrophages, which is evidence of its potential in preventing atherosclerotic CVD [35].

The aqueous extract of cinnamon increases the expression of PPARα and PPARγ and their target genes such as CD36 and GLUT4 in adipocytes [36]. It has been found that 2-methoxycinnamaldehyde, a cinnamon active ingredient, reduces the VCAM-1 expression in tumor necrosis factor alpha (TNF𝛼)-activated human umbilical vein endothelial cells (HUVECs) [37] Moreover, the cinnamophilin in cinnamon was found to be a thromboxane A2 receptor antagonist that inhibits platelet aggregation [38]. All these results highlight the beneficial effects of cinnamon in atherosclerosis.

### 2.3. Chili Peppers

Chili peppers, belonging to the genus Capsicum, are native to Mexico. *Capsicum annuum*, *Capsicum frutescens*, *Capsicum chinense*, *Capsicum pubescens* and *Capsicum baccatum* are common species of chili peppers. Chili peppers are used in most dishes and sauces such as curries, tabasco sauce, and salsa in Mexico and some other countries. In folk medicine, chili peppers are used to promote digestion and blood circulation, and exert anticancer, antioxidative and analgesic effects. Capsaicin, the major active compound of chili peppers, is used as an analgesic for muscle and joint pains. Moreover, the benefits of capsaicin on obesity, diabetes, CVD, various cancers, and dermatologic problems have been reported [39,40,41]. Intake of 4 mg of capsaicin daily significantly improved HDL levels and reduced triglyceride and C-reactive protein levels in healthy human [42].

Capsaicin modulates the functions of macrophages, ECs and VSMCs (Figure 1A,B). Kim CS et al. suggested that capsaicin inhibits NF-κB signaling in lipopolysaccharide (LPS)-stimulated peritoneal macrophages [43]. Additionally, capsaicin activates Ca^2+^/PI3K/Akt/eNOS/NO signaling to inhibit the expression of inflammatory cytokines and adhesion molecules in ECs [44]. Capsaicin activates the transient receptor potential cation channel subfamily V member 1 (TRPV1) to induce autophagy and inhibit foam cell transformation from VSMCs via the regulation of the AMP-activated protein kinase (AMPK) pathway, and enhance cholesterol efflux as well as reduce cholesterol uptake via the regulation of ABCA1 and LDL-related protein 1 expression, and thereon ameliorate atherosclerosis in ApoE^−/−^ mice fed an HFD [45,46] (Figure 1D). Dihydrocapsaicin (DHC), another chili pepper extract, was proven to reduce plaque formation through the PPARγ/LXRa pathway in ApoE^−/−^ mice fed an HFD [47]. These results demonstrate the promising effects of chili peppers in atherosclerosis.

### 2.4. Garlic (Allium sativum)

Around 80% of garlic, which is native to Central Asia, is cultivated in China. Garlic is essential in Middle Eastern and Asian cooking. In folk medicine, garlic is used to treat conditions of the digestive system and respiratory system, as well as the flu. It also offers antioxidative effects in rat models [48]. However, a systemic review indicated that there is insufficient evidence to support the effects of garlic on cardiovascular mortality and morbidity in hypertensive patients [49].

Garlic repressed neointima formation and the accumulation of cholesterol, triglycerides and phospholipids in cholesterol-fed rabbits [50]. The use of garlic extract reduced the levels of plasma total cholesterol and triglycerides in ApoE^−/−^ mice fed an HFD. Moreover, the populations of CD11b^+^ were reduced in the garlic extract group compared to the placebo group. In hyperlipidemia patients, treatment with a garlic tablet significantly reduced cholesterol and LDL-cholesterol (LDL-C) levels and increased the HDL-C level [51]. These data indicate that garlic may inhibit inflammation and lipid accumulation in the early stage of atherosclerosis [52].

Allicin, an important compound of garlic, exhibits anti-atherosclerotic effects. Lin et al. showed that allicin represses the lipid accumulation but increases ABCA1 expression and cholesterol efflux in THP-1 macrophage via the regulation of PPARγ/LXRα signaling [53]. Moreover, allicin inhibits CD36 expression through the regulation of the PPARγ pathway and prevents the differentiation of monocytes into macrophages during oxLDL stimulation [54]. These in vitro effects of garlic should be further verified in an in vivo atherosclerosis model.

### 2.5. Ginger (Zingiber officinale)

Ginger, native to South Asia, is used as a spice in cooking and in traditional medicine. It is widely cultivated in India, China, and other South Asian countries. Ginger is a commonly used for seafood, meat, and vegetarian dishes in Indian, Chinese, Korean, Japanese, and other South Asian recipes. Ginger is currently used to treat colds, headaches, inflammation, cancer and type 2 diabetes [55].

Ginger has been shown to exert anti-inflammatory, antioxidative, anti-platelet, and other anti-atherosclerotic effects in cell and animal studies [56]. Ginger inhibited lipid peroxidation without lowering the blood lipid level in cholesterol-fed rabbits, indicating that the protection ginger offers against atherosclerosis could be attributed to free-radical scavenging [57]. The major component of ginger—6-gingerol—inhibits PI3K/AKT/mTOR signaling and increases Beclin1 expression to promote autophagy in HUVECs [58]. Kamato D et al. demonstrated that (S)-6-gingerol inhibits the synthesis of transforming growth factor (TGF)-β-stimulated proteoglycan in VSMCs, indicating the role of ginger in the prevention of atherosclerotic CVD [59]. Moreover, the intake of a ginger extract diet ameliorated the lipoprotein profile in hamsters through the enhancement of hepatic cholesterol-7α-hydroxylase (CYP7A1) activity, and reductions in the mRNA levels of intestinal cholesterol absorption proteins, including NPC1L1, ACAT2, and MTP [60]. Such evidence elucidates the potential of ginger in the prevention of atherosclerosis. 

### 2.6. Anise (Pimpinella anisum)

Anise, a flowering plant, is native to the Mediterranean region and Southwest Asia. It is used as a spice in cooking and flavoring in alcohol, as well as in essential oils. In traditional Iranian and Turkish medicine, anise seeds are used for their analgesic, disinfectant and diuretic properties, and to increase milk production [61].

There is no in vivo evidence or human studies regarding the effects of anise on atherosclerosis. The anise extracts—flavonoids and phenolic acids—have potential antioxidative effects and enhance free-radical scavenging activity [62]. The ethanolic extract of anise inhibited LPS-induced NO production in RAW 264.7 macrophages, indicating the potential of anise in oxidative damage protection and its anti-inflammatory effects [63]. Accordingly, anise may reduce vascular inflammation, which is relevant to the process of atherosclerosis.

### 2.7. Chinese toon (Toona sinensis)

Chinese toon is a spice native to eastern and south-eastern Asia. It has antioxidative, anti-inflammatory, anticancer, and anti-hyperglycemic effects, and promotes digestion.

Chinese toon extracts increased the resistance of oxidative stress, but decreased prostaglandin I2 (PGI2) and interleukin (IL)-1 production in 2,2′-Azobis(2-amidinopropane) dihydrochloride (AAPH)-treated HUVECs. These extracts inhibit the expression of adhesion molecules VCAM-1, ICAM-1 and E-selectin in free radical-damaged ECs [64]. Although the effects of Chinese toon on atherosclerosis in human studies are lacking, Chinese toon has been showed to inhibit the release of inflammatory cytokines, IL-1β and TNF-α, via downregulation of NF-κB signaling in LPS-induced NF-κB activation transgenic mice [65]. Accordingly, Chinese toon exerts atheroprotective effects via its anti-inflammatory and oxidative stress reduction properties.

### 2.8. Clove (Syzygium aromaticum)

Clove, which originated in Indonesia, is used as a spice in Asian, African, and Middle Eastern countries. It is mainly produced in India, Malaysia, Sri Lanka, Madagascar, Pakistan and Tanzania. In folk medicine, cloves are used for their potential anti-bacterial, anticancer, anti-inflammatory, and blood sugar regulation effects [66].

The effects of cloves on atherosclerosis in human studies was unclear; however, cloves not only reduced the blood glucose and lipid levels in type 2 diabetic rats fed an HFD, but also improved the rate of hepatic injury via its antioxidative effects [67]. Clove extracts significantly reduced the release of pro-inflammatory cytokines, IL-1β, IL-6 and IL-10, in LPS-induced mouse peritoneal macrophages, possibly through the regulation of NF-κB signaling [68]. Further studies are necessary to further clarify the mechanisms, and confirm the benefits associated with cloves in atherosclerosis.

### 2.9. Coriander (Coriandrum sativum)

Coriander, native to the Mediterranean region, is grown in regions extending from western Asia to southern Europe. Coriander seeds, which have a bittersweet and spicy taste, are used in curry powders, while the leaves are used to make sauces. In human traditional medicine, coriander is used for its antioxidative, anti-inflammatory, anti-hypertensive, anticancer, and anti-dyslipidemic properties [69,70].

Coriander extract was found to prevent oxidative damage from isoproterenol-induced ROS production in male Wistar rats [71]. Orally-fed aqueous coriander extract reduced the levels of plasma glucose, insulin, total cholesterol, LDL-C and triglycerides in rats fed with HFD [72]. Moreover, coriander seed extract prevented LDL oxidation in RAW 264.7 macrophages and inhibited the production of total cholesterol, very low-density lipoprotein (VLDL) and triglycerides, and plaque formation in atherosclerotic rats [73]. Coriander extracts obtained from the plant’s stems and leaves inhibit nitric oxide and prostaglandin E2 production in LPS-stimulated RAW 264.7 macrophages. Coriander extract also repressed the affinity of LPS-induced NF-κB nuclear protein–DNA binding affinity and expression of phosphorylated MAPKs in macrophages [74]. Hence, coriander might ameliorate atherosclerosis through the repression of oxidative damage and inflammatory signaling, and modification of the lipid profile.

### 2.10. Dill (Anethum graveolens)

Dill, widely cultivated in Eurasia, is used as a spice in cooking. In Europe, dill is commonly used to prepare soup, butter, salad dressing and fish seasoning. In China and Middle Eastern countries, it is used to cook dumplings and kebabs. In folk medicine, the spice is used for the prevention of digestive and respiratory disease, and to reduce blood cholesterol and glucose levels [75].

Dill is used in traditional Asian medicine for the control of diabetes and cardiovascular disorders. It was found that the levels of serum triglycerides, LDL, VLDL, and blood glucose significantly reduced in dill extract-treated hamsters fed a high-cholesterol diet, compared to the sham group. Moreover, dill extract was found to exhibit inhibitory effects of HMG-CoA reductase activity in high cholesterol fed hamsters [76]. However, clinical human studies revealed that treatment with dill only mildly reduced the total cholesterol and triglyceride levels, with no effects on HDL or LDL levels in hyperlipidemic patients [51]. Further studies are necessary to verify the effects of dill on atherosclerotic CVD and metabolic syndrome.

### 2.11. Rosemary (Rosmarinus officinalis)

Rosemary is native to the Mediterranean region, and its stems, leaves and flowers can be extracted for medicinal, culinary and ornamental purposes. As it is particularly drought-tolerant, rosemary is commercially grown and cultivated in countries across the world. In folk medicine, rosemary is used to eliminate bloating, enhance memory, relieve headache symptoms and reduce hair loss; it also has a good effect in alcohol hangovers, dizziness, and tension headaches [77]. Although human studies were lacking, rosemary has been found to have anti-atherogenic [78] and anti-hypertensive effects in mice [79], lipid-lowering effects in rabbits [80], antioxidative effects in rats, anti-inflammatory effects in rats [81], anti-depressive effects in mice [82] and anti-bacterial effects in cell-line studies [83].

Rosemary extracts include carnosic acid, carnosol, rosmanol, ursolic acid (UA), betulinic acid and rosmarinic acid (RA), all of which are pharmacological constituents. UA inhibited the development of atherosclerotic lesions, and suppressed monocyte transmigration in atherogenesis-prone LDL receptor knockout mice (*ldlr*^−/−^) fed an HFD [78]. Moreover, UA inhibited macrophage recruitment in *ldlr*^−/−^ mice fed an HFD. Rosmanol and epirosmanol, which are extracted from rosemary, was found to inhibit LDL oxidation in human blood [84]. These results suggest that rosemary has the potential to prevent and treat atherosclerotic CVD.

Tu et al. demonstrated that the use of rosemary extract activated the AMPK and PPAR-γ signaling pathways for the regulation of glucose and lipid metabolism in HepG2 cells [85]. Furthermore, UA inhibited the phosphorylation of MAPKs (ERK and c-Jun N-terminal kinase, JNK) and reduced the signaling of NF-κB and activator protein-1 (AP-1) in lymphocytes. It also suppressed the activation, proliferation and cytokine secretion in T cells, B cells and macrophages, indicating its potential anti-inflammatory effects [86]. Therefore, conducting studies focusing on these signaling pathways may be indispensable to gain knowledge on how rosemary improves atherosclerosis.

### 2.12. Saffron (Crocus sativus)

Saffron was first cultivated in Greece and the areas surrounding it, extending from the Western Mediterranean region to North Africa, North America and Oceania. More than 90% of the world’s saffron is produced in Iran [87]. Saffron has been used as a spice and dye, and also has medicinal properties. It is a highly-valued medicinal plant that is traditionally used for its analgesic and anticancer properties, as well as for menstrual regulation and hepatic protection [88].

A natural carotenoid obtained from saffron inhibited glycation and oxidation product formation, plaque formation and inflammation in diabetic-atherosclerotic rats with reductions in the blood triglyceride, LDL and glucose levels [89]. In addition, in an investigation of 150 metabolic syndrome patients taking saffron or placebo randomly, patients who took saffron had lower antibody titres to heat-shock proteins 27, 60, 65 and 70, indicating the anti-inflammatory properties of the spice [90]. The use of saffron extract was found to not only reduce IL-6, TNF-αand monocyte chemoattractant protein-1 levels, but also glucose and triglyceride levels. In ApoE^−/−^ mice fed an HFD, saffron promoted plaque stabilization by increasing the production of collagen, elastin, and SMCs, and reducing the macrophage content [91]. The use of an aqueous extract of saffron not only modulated mir21 and mir142-3p in HUVECs, but also inhibited ROS which regulated the NF-κB pathway indirectly [92]. The active ingredient of saffron—crocin—promoted M2 macrophage polarization and decreased the total cholesterol and LDL-C levels, but increased the HDL-C levels in an atherosclerosis rat model, possibly through the inhibition of the NF-κB pathway [93]. These results suggest that saffron is a potential therapeutic agent for atherosclerosis.

### 2.13. Star anise (Illicium verum)

Star anise is native to Vietnam and China. After drying, star anise has a slightly sweet taste and is used as a unique flavoring agent. Star anise can be used in wines, soaps, body lotions, and essential oils. In Chinese folk medicine, star anise is used to treat stomach aches, kidney dysfunction, vomiting, the flu and insomnia. It has antimicrobial, insecticidal, antioxidative, analgesic, sedative and convulsive effects in pharmacology [94]. Shikimic acid, extracted from star anise, is a precursor of oseltamivir which is an anti-viral medication used to treat influenza [95]. 

In Asia, star anise is used to treat inflammatory diseases such as skin inflammation, rheumatism, and bronchitis. It inhibited the ICAM-1 expression and JAK/STAT signaling in interferon-γ-induced HaCaT cells [96]. Moreover, the essential oil of star anise repressed NO and PGE2 expression in LPS-induced RAW 264.7 cells [97]. Star anise inhibits NF-κB transcriptional activity in TNF-α-stimulated human aortic smooth muscle cells (HASMCs). In ApoE^−/−^ mice fed an HFD, star anise improved the lipid profiles, and ameliorated atherosclerosis. The levels of inflammatory cytokines, such as TNF-α and IL-1β, were reduced after star anise treatment [98]. These studies highlight the potential effects of star anise in treating atherosclerotic CVD.

### 2.14. Tarragon (Artemisia dracunculus)

Tarragon, characterized by its bittersweet flavor, is commonly used in Mediterranean cuisine. It is mainly cultivated in southern Europe, Russia, and the United States. In folk medicine, tarragon is used to treat conditions of the digestive system. In traditional Russia therapy, tarragon is used for its anti-inflammatory properties, and also has wound-healing, antiulcer, anticancer and anti-bacterial effects [99].

Tarragon extract has been proven to exert anticoagulation activity with inhibitory effects on platelet adhesion, aggregation and secretion in healthy human platelets [100]. A herbal mixture containing this spice improved the blood total cholesterol, LDL, HDL, triglyceride, and glucose levels as well as reduced the prevalence of atherosclerotic lesions and lipid accumulation in the liver in ApoE^−/−^ mice fed a Western diet [101]. Even though further evaluation of the underlying molecular mechanisms is needed, the anti-coagulant and anti-atherogenic potential of tarragon indicates its potential in treating atherosclerotic CVD.

### 2.15. Turmeric (Curcuma longa)

Turmeric, native to India and Southeast Asia, is commonly used as food coloring in Asian dishes, such as curries. In folk medicine, turmeric is used for gynecological problems, stomach diseases, liver diseases, infectious diseases, and blood diseases. It has been suggested that turmeric has the potential to treat proinflammatory diseases, cancer, diabetes, obesity, and atherosclerosis [102].

Zikaki et al., indicated that curcumin induced H9c2 cell apoptosis via the upregulation of ROS and JNKs [103]. Recent evidence suggests that curcumin not only scavenges free radicals, but also activates several transcription factors such as PPARγ, and LXRα in mouse macrophages [104]. Curcumin upregulates ABCA1 expression and cholesterol efflux via AMPK-SIRT1-LXRα signaling in macrophages [105]. Moreover, curcumin represses ERK, JNK, p38 and NF-κB signaling, inhibits M1 macrophage polarization and reduces the expression of inflammatory cytokines, TNF-α, IL-6, and IL-12B (p40), in THP-1 macrophages [106]. Although further in vivo and human studies are necessary, these studies demonstrate the anti-inflammatory and atheroprotective effects of curcumin.

## 3. Recommendations for Diet Adjustment in Prevention of CVD

Vascular inflammation and lipid accumulation are the main components of atherosclerosis. Chili peppers, clove, coriander, and saffron suppress NF-κB signaling and further inhibit the release of pro-inflammatory cytokines in macrophages (Figure 1A). Rosemary, turmeric and coriander inhibit the inflammatory reaction in macrophages via the downregulation of MAPKs. Moreover, black pepper, chili pepper, garlic and turmeric activate PPAR and LXR signaling and increase ABCA1 expression and cholesterol efflux in macrophages. In ECs, star anise and Chinese toon inhibit the expression of adhesion molecules, ICAM-1 and VCAM-1, and the expression of pro-inflammatory cytokines, IL-1β and TNFα, via the suppression of NF-κB signaling (Figure 1B). Moreover, black pepper inhibits VSMC proliferation through repressing MAPK signaling (Figure 1C). These effects contribute to the atheroprotective benefits of those spices. Regarding in vivo atheroprotective effects, chili pepper triggered TRPV1 activation to reduce vascular lipid accumulation and attenuate atherosclerosis in ApoE^−/−^ mice fed an HFD [45] (Figure 1D). Additionally, some spices, including saffron, coriander, star anise, tarragon and garlic, have been shown to provide atheroprotective effects in atherogenesis-prone rodents and rabbits [50,73,89,98,101]. All these results provide strong evidence indicating the anti-atherosclerotic properties of spices.

There are some human studies evaluating the effects of spices, including saffron, turmeric, chili pepper, garlic, and cinnamon, on hypertension. Although saffron, turmeric, and chili pepper had neutral effects on blood pressure, cinnamon demonstrated significant blood pressure lowering effects in patients with diabetes. Moreover, garlic has been shown to have the potential to reduce blood pressure in hypertensive patients [29,42,49,107,108]. These studies provide information on the beneficial roles of spices in reducing cardiovascular risk factors.

Currently, there are no available epidemiological data showing that consumption of spices is associated with reduction of cardiovascular events as well as the recommended amounts of spices to be consumed; however, from our review, spices with potential cardiovascular protective effects could be added to diets to prevent CVD. Although spices cannot replace nutrients from vegetables and fruits, for people living in areas in which there is a shortage of fruits or vegetables due to environmental factors, spices can be used in the daily diet. The Mediterranean diet, which includes the use of a host of spices in addition to olive oil, is healthier than other diets. Black pepper, cinnamon, chili pepper, garlic and ginger are commonly used in different diets across cultures, and this inhibits atherosclerosis via different signaling pathways (Table 2). Such spices can be applied to the Western diet, which is characterized by high levels of fat, salt and sugar, to prepare food, for pickling, to cook meat, and increase the flavor of dishes, to derive the benefits of their atheroprotective properties. Furthermore, cinnamon, with its inhibitory effects on atherosclerosis, can be used to prepare desserts. In Chinese cooking, although chili pepper, garlic and ginger are used widely, the use of Chinese toon and star anise can be considered to make sauces and seasonings, as they have anti-inflammatory and atheroprotective effects. Metabolic syndrome and diabetes are highly prevalent in Middle Eastern countries due to the high meat consumption and low intake of vegetables and fruits [10]. In such cases, the use of turmeric, coriander and other common spices in commonly consumed dishes may reduce the risk of atherosclerotic CVD. While it may be difficult to change dietary habits, nutritionists must be encouraged to formulate appropriate food mixtures containing a larger proportion of beneficial ingredients as a means to promote cardiovascular health, worldwide.

Most of the studies reviewed in this article focused on the in vitro effects of the spices relevant to atherosclerosis. The in vivo effects of spices, such as rosemary, black pepper, ginger and turmeric, on atherosclerosis have not yet been established. Some spices, such as tarragon and dill, have beneficial roles in lipid metabolism with unknown regulatory mechanisms. Furthermore, it is critical to understand the interactions between spices, and those between spices and other foods through extensive study. Finally, recent studies have demonstrated that the transportation role of exosome or extracellular vesicles, inflammasome activation, and neutrophil extracellular trap (NET) activation/NETosis are some novel mechanisms involved in the process of atherosclerosis [109,110,111]. It may prove useful to perform studies focusing on the effects of spices on these mechanisms to help establish the benefits of spices on the prevention and treatment of atherosclerotic CVD. There is also a need for further human observational and clinical studies to investigate the association between atherosclerosis and spices.

## 4. Conclusions

In this review, we demonstrated the potential atheroprotective effects of several spices through the inhibition of inflammatory reaction and lipid accumulation. The CVD prevention benefit associated with the consumption of some of these spices can be demonstrated both in vitro and in vivo, although the atheroprotective effects of some spices were only demonstrated in the cell culture system. Based on the promising in vivo and in vitro effects of spices on atherosclerosis, we are confident that spices will be used for the treatment of patients with atherosclerotic CVD in the near future, either as new therapeutic agents or as adjuvants with current medications. Nutritionists and consumers must focus on adjusting food recipes and modifying dietary habits, respectively, to avoid CVD development in the long run. The use of spices with atheroprotective properties must be encouraged across countries to promote general health.

Although further in vivo investigations and human studies are necessary to prove the beneficial effects of various spices on atherosclerosis, our review suggests that the modification of dietary habits through the use of healthy spices in meals may reduce the risk of atherosclerotic CVD.

## Figures and Tables

**Figure 1 nutrients-10-01724-f001:**
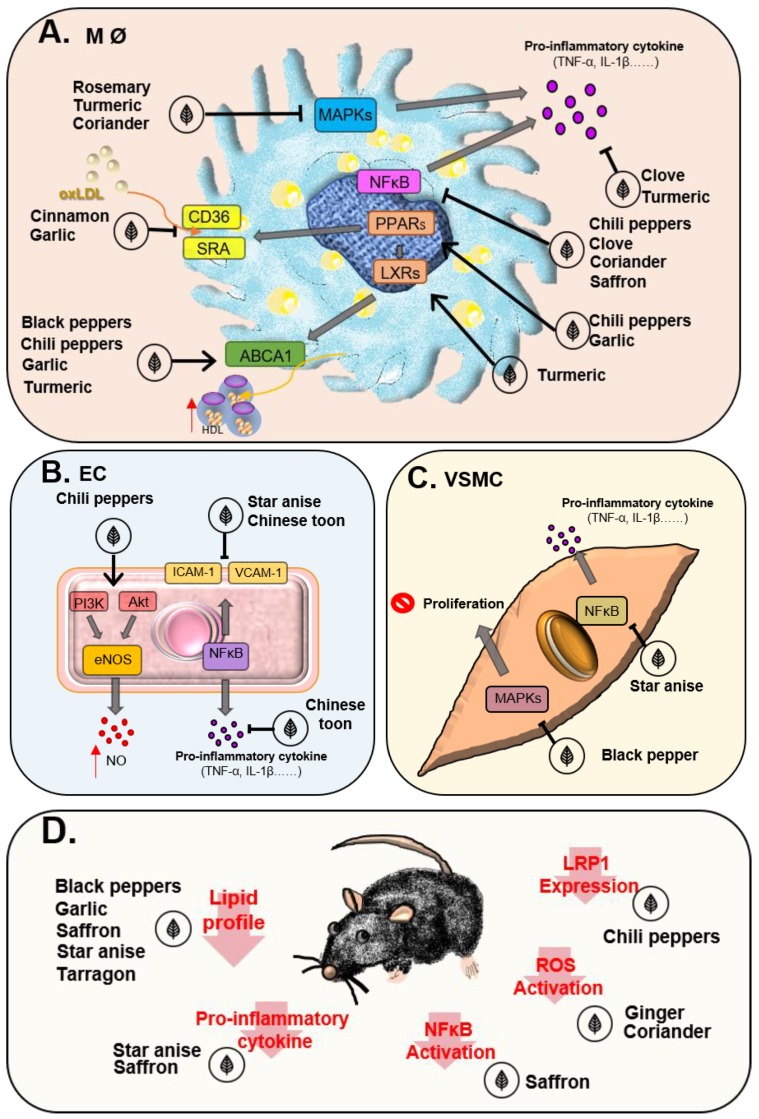
The atheroprotective effects of spices in vitro and in vivo. The atheroprotective effects of spices in macrophages (**A**), endothelial cells (**B**), smooth muscle cells (**C**) and animal model (**D**).

**Table 1 nutrients-10-01724-t001:** Spices in different culture diets.

Diet Category	Spices	Prevalence of CVD	Reference
Western diet	Spice-free with salt and sugar	11–15%	[21]
Mediterranean diet	Anise, basil, bay, cardamom, cinnamon, chervil, chilis, chives, cloves, cumin, coriander, dill, fennel, fenugreek, garlic, mace, marjoram, mint, nutmeg, oregano, peppers, rosemary, saffron, sage, savoury, sumac, tarragon and thyme.	1.5–3.2%	[3,4,5,6,22,23,24]
Chinese diet	Cardamom, cinnamon, cumin, cloves, peppers, nutmeg, peppercorns, fennel, star anise, garlic, ginger, peppers and chili peppers.	5%	[7,8,21]
Indian diet	Cardamom, clove, cassia, peppers, cumin, coriander, nutmeg, mustard seed, fenugreek, turmeric, saffron and garlic.	7–11%	[9,21]
Arabic diet	Saffron, peppers, allspice, turmeric, garlic, cumin, cinnamon, parsley, and coriander.	7–12%	[10,21]

**Table 2 nutrients-10-01724-t002:** Spices and its potential mechanism in atherosclerosis improvement.

Spices	Place of Origin	Extracts	Atheroprotective Effects	Potential Mechanism	References
Commonly used spices					
Black pepper	South India	Piperine	Anti-oxidation Anti-atherogenesis	(−) Lipid profile, including total cholesterol, LDL, and triglycerides (−) VSMCs proliferation via repressing pERK1/2 (−) ROS production and pp38 expression (+) ABCA1 expression	[20,21,22,23,24,25,26,27,28]
Cinnamon	India and Sri Lanka	Cinnamaldehyde 2-Methoxycinnamaldehyde Cinnamophilin	Anti-coagulation Anti-oxidation Anti-inflammation Anti-diabetes	(−) Phagocytosis of LDL (−) CD36 and SRA expression (−) TNFα-activated VCAM-1 expression (−) Platelet aggregation	[29,30,31,32,33,34,35,36,37,38]
Chili peppers	Mexico	Capsaicin Dihydrocapsaicin	Anti-oxidation Anti-inflammation Anti-atherogenesis	(−) IκBα degradation and NF-κB pathway (−) Plaque formation via PPARγ/LXRα pathway (+) LXRα and ABCA1 expression (+) Autophagy to inhibit form cell formation	[39,40,41,42,43,44,45,46,47]
Garlic	Central Asia	Allicin	Anti-oxidation CVD protection Anti-atherogenesis	(−) Lipid profile, including total cholesterol, LDL, and triglycerides (−) Inflammation and lipid accumulation in early stage of atherosclerosis (+) ABCA1 expression and cholesterol efflux through PPARγ/LXRα signalling	[48,49,50,51,52,53,54]
Ginger	Southern Asia	6-gingerol	Anti-inflammation Anti-oxidation	(−) Lipid peroxidation (−) PI3K/AKT/mTOR signalling (+) Beclin1 expression to promote autophagy	[55,56,57,58,59,60]
Area specific spices					
Anise	Mediterranean region and Southwest Asia	Flavonoids Phenolic acids	Anti-oxidation Free radical scavenging activity	(−) NO production (+) Free radical scavenging	[61,62,63]
Chinese toon	Eastern and South-eastern Asia	N/A	Anti-oxidation Anti-inflammation Lipid-lowing effects	(−) VCAM-1, ICAM-1 and E-selectin (−) LPS-induced IL-1β and TNFα expression	[64,65]
Clove	Indonesia	N/A	Anti-inflammation Anti-oxidation Reduction of hyperglycemia and hyperlipidemia	(−) LPS-induced IL-1β and IL-6 expression through NF-kB pathway	[66,67,68]
Coriander	Mediterranean region	N/A	Anti-oxidation Anti-inflammation Anti-dyslipidemia	(−) LDL oxidation (−) Isoproterenol-induced ROS production (−) Total cholesterol, VLDL, triglyceride and plaque formation (−) LPS-stimulated nitric oxide and PGE2 production through NF-κB and MAPKs activation	[69,70,71,72,73,74]
Dill	Eurasia	N/A	Cholesterol and glucose level management	(−) HMG-CoA reductase activity (−) Total cholesterol and triglyceride	[51,75,76]
Rosemary	Mediterranean region	Carnosic acid Ursolic acid Rosmarinic acid Rosmanol	Anti-atherogenesis Anti-hypertension Lipid-lowing effects Anti-oxidation Anti-inflammation	(−) Monocyte transmigration (−) Macrophage recruitment (−) LDL oxidation activity (−) MAPKs activation	[77,78,79,80,81,82,83,84,85,86]
Saffron	Greece area	Carotenoid Crocin	Anti-oxidation Anti-glycation Anti-inflammation	(−) plaque formation (−) IL-6, TNF-α and MCP-1 expression (−) miR-21 and miR-142-3p expression (−) ROS response (−) Lipid formation	[87,88,89,90,91,92,93]
Star anise	Vietnam and China	Shikimic acid	Anti-oxidation Anti-inflammation	(−) ICAM-1 expression (−) LPS-induced NO/iNOS and PGE2/COX-2 expression (−) TNFα-stimulated NF-κB transcriptional activity	[94,95,96,97,98]
Tarragon	Southern Europe, Russia, and the United States	N/A	Anti-inflammation Anticoagulant activity	(−) Atherosclerotic lesion in aortic valve and lipid accumulation	[99,100,101]
Turmeric	Indian and Southeast Asia	Curcumin	Anti-inflammation Anti-oxidation Anti-atherogenesis Anti-diabetes	(−) M1 macrophage polarization and TNF-α, IL-6, and IL-12 expression through inhibition of MAPKs and NF-kB activation (+) Free radical scavenging (+) PPARγ, and LXRα transcription factors activity (+) ABCA1 expression and cholesterol efflux	[102,103,104,105,106]

(+) promote (−) repress. ABCA1, ATP-binding cassette (ABC) subfamily A; COX-2, cytochrome c oxidase subunit 2; ERK1/2, extracellular signal-regulated protein kinases 1 and 2; HMG-CoA, 3-hydroxy-3-methyl-glutaryl-coenzyme A reductase; ICAM-1, intercellular adhesion molecule 1; LPS, Lipopolysaccharides; LDL, Low-density lipoprotein; LXRα, liver X receptor alpha; MAPK, mitogen-activated protein kinase; NF-κB, nuclear factor kappa-light-chain-enhancer of activated B cells; NO, nitric oxide; PGE2, prostaglandin E2; PPARγ, peroxisome proliferator-activated receptor gamma ROS, reactive oxygen species; SRA, scavenger receptor class A; TNF-α, tumor necrosis factor alpha; VCAM-1, vascular cell adhesion molecule 1; VSMCs, vascular smooth muscle cells, N/A, not available.

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
