# Peer review of "Spices and Atherosclerosis"

_nutrients, 2018, doi:10.3390/nu10111724_

Round 1

Reviewer 1 Report

Dr. Tsui PF et al. summarized data regarding spices and atherosclerosis. Spices from all over the world were covered and data indicating their atheroprotective effects were listed. This review provides useful information to the physicians and nutritionists applicable into their daily practices. This reviewer list only following points to be addressed.

As authors mentioned, a large part of data is in vitro or animal data. To apply these results on daily dietary habits, epidemiological data showing that consumption of these spices is associated with reduction of cardiovascular events should be demonstrated. In addition, it is better to describe recommended amount of spices to be consumed if possible.

There is a possibility that some spices influence on the blood pressure. Authors should mention this point if there are human data about each spices and blood pressure.

In the Introduction (1.1. Line 35), authors should mention the place or country where the prevalence of CVD is investigated.

Author Response

Point 1: As authors mentioned, a large part of data is in vitro or animal data. To apply these results on daily dietary habits, epidemiological data showing that consumption of these spices is associated with reduction of cardiovascular events should be demonstrated. In addition, it is better to describe recommended amount of spices to be consumed if possible.

Response 1: Thank you for your valuable comments. Currently, there are no available epidemiological data showing that consumption of spices is associated with reduction of cardiovascular events as well as the recommended amounts of spices to be consumed; however,from our review, spices with potential cardiovascular protective effects could be added to diets to prevent CVD. The above description had been added in section 3, line 264. 

Point 2:There is a possibility that some spices influence on the blood pressure. Authors should mention this point if there are human data about each spices and blood pressure.

Response 2:Thank you for your valuable comments. There are some human studies evaluating the effects of spices, including saffron, turmeric, chili pepper, garlic, and cinnamon, on hypertension. Although saffron, turmeric, and chili pepper have neutral effects on blood pressure, cinnamon demonstrated significant blood pressure lowering effects in patients with diabetes. Moreover, garlic has been shown to have the potential to reduce blood pressure in hypertensive patients [1-5]. These studies provide information on the beneficial roles of spices in reducing cardiovascular risk factors. We had added the above description in section 3, line 258. 

Point 3:In the Introduction (1.1. Line 35), authors should mention the place or country where the prevalence of CVD is investigated.

Response 3:Thank you for your suggestion. The prevalence of CVD is estimated to increase from 36.9% to 40.5% from 2010 to 2030 in the United States, and the associated medical cost will increase by 200%. We had added the above description in the introduction section 1.1., line 35

Reference

1.         Akilen, R.; Tsiami, A.; Devendra, D.; Robinson, N., Glycated haemoglobin and blood pressure-lowering effect of cinnamon in multi-ethnic Type 2 diabetic patients in the UK: a randomized, placebo-controlled, double-blind clinical trial. Diabet Med 2010,27, 1159-67.

2.         Qin, Y.; Ran, L.; Wang, J.; Yu, L.; Lang, H. D.; Wang, X. L.; Mi, M. T.; Zhu, J. D., Capsaicin Supplementation Improved Risk Factors of Coronary Heart Disease in Individuals with Low HDL-C Levels. Nutrients 2017,9, 9.

3.         Stabler, S. N.; Tejani, A. M.; Huynh, F.; Fowkes, C., Garlic for the prevention of cardiovascular morbidity and mortality in hypertensive patients. Cochrane Database Syst Rev 2012, 8, CD007653.

4.         Joshi, J.; Ghaisas, S.; Vaidya, A.; Vaidya, R.; Kamat, D. V.; Bhagwat, A. N.; Bhide, S., Early human safety study of turmeric oil (Curcuma longa oil) administered orally in healthy volunteers. J Assoc Physicians India 2003,51, 1055-60.

5.         Nasim Abedimanesh, A. O., S Zahra Bathaie, Saeed Abedimanesh, Behrooz Motlagh, Mohammad Asghari Jafarabadi, and Mohammadreza Taban Sadeghi, Effects of Saffron Aqueous Extract and Its Main Constituent, Crocin, on Health-Related Quality of Life, Depression, and Sexual Desire in Coronary Artery Disease Patients: A Double-Blind, Placebo-Controlled, Randomized Clinical Trial. Iran Red Crescent Med J. 2017, 19(9):e13676.

Reviewer 2 Report

The manuscript by Pi-Fen Tsui provides a comprehensive overview of the influence of dietary nutrients such as spices towards inhibition of inflammatory reaction and lipid accumulation, indicating their potentially protective effects in atherosclerosis. The manuscript presents detailed descriptions on the epidemiological usage of spices and the application of various spices in different experimental systems (cell culture, animal) towards elucidation of their underlying molecular mechanisms that lead to their atheroprotective properties. The manuscript is generally well written.

Comments on the manuscript:

1. In the introduction, line 33, the authors mention that ‘Atherosclerotic cardiovascular disease (CVD) is one of the leading causes of death worldwide’. Given that the rest of the paragraph is referring to CVD, and numbers of estimated rise in future prevalence provided in reference 1 are referring to CVD and not atherosclerotic CVD then the above sentence is slightly misleading. It may be best if the word Atherosclerotic is removed from it.

2. In addition to diet, atherosclerosis may be caused by additional factors including genetic and environmental aspects. This should also be mentioned in the manuscript.

3. In section 2.10, treatment with dill (Anethum graveolens) in human hyperlipidaemic patients did not appear to have a significant beneficial effect in the parameters examined. The authors should therefore tone-down the concluding sentence of the paragraph regarding the benefits of dill in the prevention and treatment of atherosclerotic CVD.

4. Table 1 is very helpful and provides a clear overview of the different spices used in different geographical regions. Is there any information available regarding the prevalence of atherosclerosis in these geographical areas? While this may provide interesting information, any extrapolation regarding the beneficial effect of spices in these populations should be done with caution since additional factors may also be responsible for any differences in atherosclerosis prevalence in these areas. 

5. Figures 1-3 are very useful. However, it may be better if they are included at a latter point in the text (eg. in section 3) so as to provide a general overview of the molecular effects exerted by the various spices. Also, including them in a single figure, as separate panels, may be helpful in demonstrating the effect of spices in different cell types and animal models.    

6. While there is ample of evidence indicating the protective effects of spices, this is primarily based on cell culture studies. Some evidence is available on animal models, however no major effect was seen in a human study (reference 44). In reference 20, the study by Sivakumar reports the hypolipidaemic effect of Piper in rats. As reported in the methods of this paper, rats (weighing 90-100gr) were fed with 150mg/kg per day piper. Given that the average weight of humans is ~70kg, do the authors envision that the beneficial effects of piper could be applicable to humans? And this is relevant to all spices.     

7. At several points in the manuscript it is mentioned that spices can benefit the prevention and treatment of atherosclerotic CVD. Do the authors envision that, in the future, spices will be used for the treatment of patients with atherosclerosis? How would this fit with the various pharmacological agents (eg statins) that are currently widely used as therapeutic agents?     

Minor comments

1. Section 2.10, Line 141: Asian machine, do the authors mean Asian medicine? Please correct accordingly.

2. Section 2.10, Line 145: mild should be replaced by mildly

Author Response

Major comments 

Point 1:In the introduction, line 33, the authors mention that ‘Atherosclerotic cardiovascular disease (CVD) is one of the leading causes of death worldwide’. Given that the rest of the paragraph is referring to CVD, and numbers of estimated rise in future prevalence provided in reference 1 are referring to CVD and not atherosclerotic CVD then the above sentence is slightly misleading. It may be best if the word Atherosclerotic is removed from it.

Response 1: Thank you for your comments. This has been omitted in the revised manuscript (Section 1, line 33).

Point 2:In addition to diet, atherosclerosis may be caused by additional factors including genetic and environmental aspects. This should also be mentioned in the manuscript.

Response 2: Thank you for this suggestion. We have updated our content so that it now reads, “Atherosclerosis is the main cause of CVDs. The risk factors of atherosclerosis and CVD include diabetes, smoking, hypertension, dyslipidemia, obesity and age.(Section 1, line 37)

Point 3:In section 2.10, treatment with dill (Anethum graveolens) in human hyperlipidaemic patients did not appear to have a significant beneficial effect in the parameters examined. The authors should therefore tone-down the concluding sentence of the paragraph regarding the benefits of dill in the prevention and treatment of atherosclerotic CVD.

Response 3: Thank you for your positive comments. The content has been revised to, “Further studies are necessary to verify the effects of dill on atherosclerotic CVD and metabolic syndrome.” (Section 2.10, line 148)

Point 4: Table 1 is very helpful and provides a clear overview of the different spices used in different geographical regions. Is there any information available regarding the prevalence of atherosclerosis in these geographical areas? While this may provide interesting information, any extrapolation regarding the beneficial effect of spices in these populations should be done with caution since additional factors may also be responsible for any differences in atherosclerosis prevalence in these areas. 

Response 4: Thank you for your positive comments. The information regarding the prevalence of atherosclerosis in different geographical areas has been provided in Table 1.

Point 5:Figures 1-3 are very useful. However, it may be better if they are included at a latter point in the text (eg. in section 3) so as to provide a general overview of the molecular effects exerted by the various spices. Also, including them in a single figure, as separate panels, may be helpful in demonstrating the effect of spices in different cell types and animal models. 

Response 5: Thank you for your suggestion. The figures have been included in a single figure with separate panels and have been moved to section 3. 

Point 6:While there is ample of evidence indicating the protective effects of spices, this is primarily based on cell culture studies. Some evidence is available on animal models, however no major effect was seen in a human study (reference 44). In reference 20, the study by Sivakumar reports the hypolipidaemic effect of Piper in rats. As reported in the methods of this paper, rats (weighing 90-100gr) were fed with 150mg/kg per day piper. Given that the average weight of humans is ~70kg, do the authors envision that the beneficial effects of piper could be applicable to humans? And this is relevant to all spices.     

Response 6: Thank you for your suggestion. We have added “Although the beneficial role of black pepper on hyperlipidemia has been demonstrated in rats, the effects of black pepper on cardiometabolic disease as well as the amounts used in humans need further studies to verify.(Section 2.1, line 112)

Point 7:At several points in the manuscript it is mentioned that spices can benefit the prevention and treatment of atherosclerotic CVD.Do the authors envision that, in the future, spices will be used for the treatment of patients with atherosclerosis? How would this fit with the various pharmacological agents (eg statins) that are currently widely used as therapeutic agents?     

Response 7: Thanks for your comments. Based on the promisingin vivoand in vitroeffects of spices on atherosclerosis, we are confident that spices will be used for the treatment of patients with atherosclerotic CVD in the near future, either as new therapeutic agents or as adjuvants with current medications. This information has been added in section 4, line 301. 

Minor comments

Point 1:Section 2.10, Line 141: Asian machine, do the authors mean Asian medicine? Please correct accordingly.

Response 1:  This has been revised to “medicine” (Section 2.10, line 142)

Point 2:Section 2.10, Line 145: mild should be replaced by mildly

Response 2: This has been revised to “mildly” (Section 2.10, line 147)

Reviewer 3 Report

1. The references should be added to each diet category in “Table 1. Spices in different culture diets”.

2. For each spice described in this manuscript, authors should provide the information of epidemiological findings, in vivo study including human and animal studies, in vitro study and the mechanisms. If there is no such information is available, just state that fact. Furthermore, the author should provide more clear information regarding the effects of the spice. For example, as stated in this manuscript that "Rosemary has been found to have anti-atherogenic … effects”, were these effects found in animal study? Or human study? Or cell culture results?

3. What does this sentence mean as “For people living in areas in which there is a shortage of fruits or vegetables due to environmental factors, spices can be used in the daily diet” (line 372 to 373, page 13)? Can spice replace fruits or vegetables? Spice can always be added to diets no matter fruits or vegetables are available or not.

Author Response

Point 1:The references should be added to each diet category in “Table 1. Spices in different culture diets”.

Response 1: Thanks for your suggestion. The reference has been added to Table 1.

Point 2:For eachspice described in this manuscript, authors should provide the information of epidemiological findings, in vivo study including human and animal studies, in vitro study and the mechanisms. If there is no such information is available, just state that fact. Furthermore, the author should provide more clear information regarding the effects of the spice. For example, as stated in this manuscript that "Rosemary has been found to have anti-atherogenic … effects”, were these effects found in animal study? Or human study? Or cell culture results?

Response 2: Thank you for your valuable comments. Currently, there are no available epidemiological data showing that consumption of spices is associated with reduction of cardiovascular events. For each spice, the available data regarding in vivo studies including human and animal studies, in vitro studies and the mechanisms for each has been provided. Moreover, we carefully checked the given information of each spice to provide its clear effects and revised accordingly (Section 2.2, line 6; Section 2.3, line 36; Section 2.4, line 53; Section 2.6, line 91; Section 2.7, line 104; Section 2.8, line 114; Section 2.10, line 145 and 148; Section 2.11, line 156 and 165; Section 2.14, line 217 and 220; Section 2.15, line 231 and 235). 

Point 3:What does this sentence mean as “For people living in areas in which there is a shortage of fruits or vegetables due to environmental factors, spices can be used in the daily diet” (line 372 to 373, page 13)? Can spice replace fruits or vegetables? Spice can always be added to diets no matter fruits or vegetables are available or not.

Response 3:Thank you for your constructive comments. The description has been revised as follows, “From our review, spices with potential cardiovascular protective effects can be added to diets to prevent CVD. Although spices cannot replace nutrients from vegetables and fruits, for people living in areas in which there is a shortage of fruits or vegetables due to environmental factors, spices could be an alternative.”  These sentences have been added in section 3, line 265.

Round 2

Reviewer 3 Report

This revised paper is greatly improved. I do not have further comments.